# Adaptive Evolution as a Driving Force of the Emergence and Re-Emergence of Mosquito-Borne Viral Diseases

**DOI:** 10.3390/v14020435

**Published:** 2022-02-21

**Authors:** Xi Yu, Gong Cheng

**Affiliations:** 1Tsinghua-Peking Center for Life Sciences, School of Medicine, Tsinghua University, Beijing 100084, China; yuq16@mails.tsinghua.edu.cn; 2Institute of Infectious Diseases, Shenzhen Bay Laboratory, Shenzhen 518000, China; 3School of Life Sciences, Tsinghua University, Beijing 100084, China; 4Institute of Pathogenic Organisms, Shenzhen Center for Disease Control and Prevention, Shenzhen 518055, China

**Keywords:** mosquito-borne virus, mosquito-borne viral disease, adaptive evolution

## Abstract

Emerging and re-emerging mosquito-borne viral diseases impose a significant burden on global public health. The most common mosquito-borne viruses causing recent epidemics include flaviviruses in the family *Flaviviridae*, including Dengue virus (DENV), Zika virus (ZIKV), Japanese encephalitis virus (JEV) and West Nile virus (WNV) and *Togaviridae* viruses, such as chikungunya virus (CHIKV). Several factors may have contributed to the recent re-emergence and spread of mosquito-borne viral diseases. Among these important causes are the evolution of mosquito-borne viruses and the genetic mutations that make them more adaptive and virulent, leading to widespread epidemics. RNA viruses tend to acquire genetic diversity due to error-prone RNA-dependent RNA polymerases, thus promoting high mutation rates that support adaptation to environmental changes or host immunity. In this review, we discuss recent findings on the adaptive evolution of mosquito-borne viruses and their impact on viral infectivity, pathogenicity, vector fitness, transmissibility, epidemic potential and disease emergence.

## 1. Introduction

Emerging and re-emerging vector-borne diseases are major public health problems worldwide [1]. Today, infectious diseases are the second most common cause of death globally and the number one cause of death in developing countries, killing as many as 15 million people each year according to the World Health Organization. In the past few decades, several mosquito-borne viruses have emerged and re-emerged globally.

The most common mosquito-borne viruses causing recent epidemics are flaviviruses in the family *Flaviviridae*, including Dengue virus (DENV), Zika virus (ZIKV), Japanese encephalitis virus (JEV) and West Nile virus (WNV) and *Togaviridae* viruses, such as chikungunya virus (CHIKV) [2]. Several factors may have contributed to the recent re-emergence and spread of mosquito-borne viral diseases. Possible causes of the increase in mosquito-borne viral infections are global population growth, urbanization, lack of mosquito control measures, increased air travel and declining public health [3]. The combination of lowered herd immunity and failed vector control also opened the door to viral re-emergence [4]. Other determinants of re-emergence include transportation, environmental factors, ecological cycling of vectors, host genetics, viral evolution, human and mosquito population densities, mosquito species and vector capacity [5].

Nevertheless, the emergence or re-emergence of mosquito-borne viral diseases often involves the evolutionary adaptation of viruses to new amplification hosts or vectors. Here, we review recent findings on the adaptive evolution of mosquito-borne viruses and their impact on viral infectivity, pathogenicity, vector fitness, transmissibility, epidemic potential and disease emergence.

## 2. The Emergence and Re-Emergence of Major Mosquito-Borne Viral Diseases

Dengue fever is caused by a virus of the family *Flaviviridae*, which is spread by *Aedes* mosquitoes and is widespread in tropical and subtropical regions [6,7]. The epidemiological profile of Dengue fever is on the rise in endemic countries. The distribution of DENV has grown over the past 40 years and is now expected to infect 390 million people annually [5]. According to recent reports, DENV is the fastest-spreading arbovirus in the world [8,9]. China reported the largest Dengue epidemic in 2014 [10]. The reported age-standardized, disability-adjusted life year (DALY) rate for Dengue increased from 2007 to 2017, making it a major vector-borne disease, with a dramatic 26% increase globally [11]. This disease has shown trends of continued spread since the 1950s, with increased urbanization, globalization and unsuccessful vector control, ultimately leading to increased virus infection and transmission [12].

Zika virus was first isolated from a Rhesus monkey in the Zika forest of Uganda in 1947 [13]. The first outbreak of Zika virus disease was reported in Yap Island, Federated States of Micronesia [14]. By the end of 2013, New Caledonia had reported imported cases from French Polynesia, and cases of autochthonous transmission were reported in January 2014 [15,16]. Zika virus was reported to have caused a severe epidemic in the Americas in 2015 [17]. In the same year, ZIKV became prevalent in Brazil and then rapidly spread to more than 20 countries or territories in South and Central America. Over the next few years, the disease spread in limited countries in Africa and Southeast Asia. Since neurological complications, such as Guillain–Barre syndrome in adults and microcephaly in neonates, are caused by ZIKV infection, the ZIKV epidemic was declared an international health emergency by the World Health Organization in 2016 [17]. To date, no vaccine candidates have been approved for Zika fever.

Japanese encephalitis virus is one of the leading causes of viral meningoencephalitis worldwide and is mainly transmitted by the *Culex* mosquito [18]. Pigs are considered the main host for viral maintenance or expansion, while various birds are its natural reservoirs. Human JEV infection usually results in mild febrile illness, but approximately 1% of infected individuals develop encephalitis, with a mortality rate approaching 30% [19]. The World Health Organization estimates the number of Japanese encephalitis cases globally at 68,000 cases per year. Humans are dead-end hosts for JEV due to insufficient viremia to establish infection in mosquitoes [20]. Despite the successful vaccination and eradication programs in some economically affluent Asian countries, the distribution of JEV in Southeast Asia continues to grow. Recently, the Tiwi Islands in the Northern Territory of Australia reported its first JEV case in 2021 [21], while cases of indigenous JEV infection were detected in Angola in 2016 [22], raising concerns about future outbreaks in these regions.

West Nile virus is mainly transmitted by the *Culex* species mosquito. In contrast to most arboviruses, which are partially or fully host specific, WNV is known to have a wide range of mosquito vectors and hosts, including birds, horses and other mammals. West Nile virus was first isolated in 1937 from a febrile patient in the West Nile district of northern Uganda. The epidemiology and ecology of the disease were first described in the early 1950s and early 1960s, when subsequent epidemics occurred in the Mediterranean basin [23]. West Nile virus reached the Western Hemisphere in 1999 and quickly became one of the most widespread arboviruses globally [24]. Studies have shown that it has become endemic in several regions in Africa, Europe, Asia, Australia and the Middle East [25].

Chikungunya is a mosquito-borne viral disease caused by an alphavirus of the *Togaviridae* family [26]. The origin of CHIKV appears to be controversial, as some reports indicate that it was first isolated from the serum of a febrile human in Tanganyika in 1953 and others from patients suffering from fever, severe joint pain and skin rash in Uganda in 1959 [26]. Chikungunya evolved rapidly after its successful introduction into new localities and the acquisition of new anthropophilic vectors. The recurrence of CHIKV is a serious public health concern since the virus has been associated with several outbreaks in Africa, Asia and India [27]. Previous studies have confirmed that Chikungunya virus can be attributed to the disseminated epidemic in the Indian Ocean that originated in the coast of Kenya in 2004 [28]. Sporadic chikungunya epidemics were detected in Italy in 2007 and 2017 [29]. The virus reached the Americas in late 2013 and spread rapidly to surrounding countries. By 2017, it had caused more than 1.8 million suspected cases in 44 different countries [30].

## 3. Adaptive Evolution Drives Mosquito-Borne Viral Emergence and Re-Emergence

An important cause for the re-emergence of viruses is the adaptive evolution and genetic mutations that make them more virulent and allow for widespread epidemics. RNA viruses tend to acquire genetic diversity due to error-prone RNA-dependent RNA polymerases [31]. This promotes high mutation rates and recombination that support adaptation to environmental changes or host immunity [32].

### 3.1. Dengue Virus

More sequence and genetic diversity information is available for DENV than other arboviruses, possibly due to its greater impact on human health [33,34,35,36,37]. DENV consists of four major serotypes. Although a fifth serotype was recently reported [38], it has not been confirmed, and little information is available related to the evolution of DENV. The current DENV serogroup progenitors are estimated to have emerged approximately 1000 years ago using molecular clock techniques [39]. Most phylogenies indicate that DENV-4 is the most distinct serotype, followed by DENV-2; DENV-1 and DENV-3 are the most closely related serotypes [39,40,41]. Phylogenetic analysis of sylvatic and endemic/epidemic strains indicated that each serotype arose from a sylvatic ancestor [42], with this emergence estimated to have occurred approximately 125–320 years ago, depending on the serotype [39]. Based on the sequence of the intact envelope (E) gene or the E-nonstructural protein (NS) 1 border, DENV-1 is currently divided into four to five genotypes, including a sylvatic clade [43,44,45]. DENV-2 is divided into six subtypes: Sylvatic, American, Cosmopolitan, Asian 1, Asian 2 and Asian–American [40,42,43,46]. The two Asian subtypes have sometimes been grouped together into only one Asian genotype [47]. DENV-3 has been divided into four genotypes (I–IV) [46,48,49], sometimes including a V genotype [43]. Finally, DENV4 is divided into two endemic genotypes (I-II) and one sylvatic genotype and exhibits the lowest genetic diversity among serotypes [42,43,46,50]. Overall, these genotype structures may be modified as further sequences become available.

Differences in disease severity associated with individual serotypes or specific serotype sequences have been observed, and it remains unknown whether some serotypes are inherently more pathogenic than others. DENV-2 viruses are most frequently associated with Dengue hemorrhagic fever/Dengue shock syndrome (DHF/DSS) [51,52,53,54,55,56], followed by DENV-1 and DENV-3 viruses [51,57,58,59]. DENV-4 appears to be the most clinically benign, although it can also lead to serious illness [54]. A leading theory for the pathogenesis of DHF/DSS is that a higher incidence of DHF/DSS in DENV secondary infections is due to the antibody-dependent enhancement (ADE) phenomenon [60]. Antibodies from the primary infection still cross-react with other DENV serotypes after an initial phase of cross-reactive protection, but decay to non-neutralizing levels. These non-neutralizing antibodies may increase viral Fc receptor-mediated uptake by monocytes/macrophages and increase viral replication and immune activation with cytokine release [60]. As secondary infections, DENV-2 and DENV-4 have been reported to cause increased disease severity, while DENV-1 and DENV-3 appear to be more pathogenic in primary infections [51,54,58,61]. Southeast Asia appears to be a source of viral diversity, producing multiple strains, some of which are inherently more virulent than others. This is evidenced by their global spread and possible replacement of earlier DENV strains. Strong evidence from phylogenetic studies suggests that only DENV-2 strains derived from Southeast Asia are related to DHF/DSS in the Americas, not native American strains originally imported from the South Pacific [43,62]. Subsequent functional analysis also showed that the Thai DENV strains (Asian genotype) replicated to higher titers in human monocyte-derived macrophages and dendritic cells than the American genotype DENV-2 strain [63,64]. The DENV-3 serotype provides another compelling example of how increased viral diversity can lead to the emergence or evolution of viral clades closely related to DHF/DSS. Genotype III DENV-3 includes isolates from East Africa, South Asia and Latin America and is associated with increased DHF/DSS in these regions [43,49]. The emergence of an epidemic of DHF/DSS in Sri Lanka in 1989 led investigators to question the causes of this phenomenon. After excluding the possibility of a general increase in virus transmission or changes in serotype prevalence on the island, clade substitution became the decisive factor [49,59]. Both DENV-3 III subtypes A and B emerged in Sri Lanka in 1989, but only subtype B persisted after 1989 and was involved in all subsequent DHF/DSS cases on the island. In addition, subtype B DENV-3 III has since spread to the Americas, where it has also been associated with DHF/DSS epidemics [43,49]. Since viral strains with increased virulence have been identified through a combination of phylogenetic and epidemiological analyses, the next challenge will be to determine the molecular basis of this increased pathogenesis.

Several studies have been conducted to infer the role of viral molecular evolution in driving epidemics. Comprehensive sequencing of Asian and American genotypes revealed several key nucleotide differences, particularly at position 390 in protein E and in the 5′ and 3′ untranslated regions (UTRs) [65]. Substitution of N390 in the Asian genotype with the American genotype D390 reduced viral production of macrophage and dendritic cell cultures derived from human monocytes, and this reduction was enhanced by replacing the Asian genotype 5′ and 3 UTRs with those of the American genotype [63,64]. Asian DENV-2 strains are also transmitted in a larger percentage of field-caught *Aedes aegypti* than American DENV-2 strains [66,67]. Therefore, the success of the Southeast Asian DENV-2 strain may be due in part to more efficient replication in human target cells and increased transmission by vector mosquitoes. Another recent example involves clade substitution in the DENV-2 Asian American genotype identified by phylogenetic analysis of the full-length genomes of Nicaraguan patients. In Managua (Nicaragua), the severity of Dengue infection was found to be related to the replacement of the predecessor DENV-2 NI-1 clade by the NI-2B clade. The circulating NI-1 clade originated in Asia/America, with substitutions in capsid R97K, NS1-K94R and NS3 P245T. A single mutation at N245S in NS4B led to the evolution of NI-1 into the NI-2 clade. Another five mutations, M492V in E; L279F in NS1; and K200Q, T290I and R401K in NS5, have been shown to drive NI-2 into the more infectious strain NI-2B [68]. Additional studies reported that severe DHF cases in the DENV-2 outbreak in Taiwan increased between 2001 and 2002. Genomic analysis between the two outbreaks within a year revealed five nucleotide changes in the E, NS1, NS4A and NS5 genes, indicating molecular evolution of the virus [69].

Although increased viral virulence must be considered in the context of host immunity, it is still possible that more virulent Dengue viruses will continue to evolve in Southeast Asia and spread worldwide, replacing more benign genotypes [35,41]. On the other hand, some spontaneous mutations may render DENV less dominant in the host. A study investigating phylogenetic events that occurred during the 1971 South Pacific DENV-2 outbreak found that substitutions in prM and the nonstructural genes NS2A and NS4A resulted in attenuated infection [70].

### 3.2. Zika Virus

Phylogenetic studies have revealed that ZIKV has evolved into African and Asian/American clusters [71,72]. Outbreaks in Micronesia, the South Pacific islands and the Americas were caused by the Asian/American clade [71]. Mutations arising in the ZIKV Asian/American lineage have been shown to assist the virus in its acquisition by vectors as well as adaptation to hosts. The NS1 of many flaviviruses, including DENV and ZIKV, is secreted by host cells and is present in host blood in large quantities during acute infection [73,74,75,76]. In addition to assisting in the pathogenesis of flaviviruses in the host, when acquired with virions, NS1 enhances viral infectivity in mosquitoes by overcoming the mosquito midgut immune barrier [74]. This feature of NS1 may increase the likelihood of mosquitoes acquiring the virus during brief periods of viremia and viral epidemics in nature, and it may be a survival strategy by which arboviruses have evolved to cycle efficiently between two distinct host environments. Therefore, viral genetic mutations affecting NS1 secretion may affect viral transmission from vector to host and/or vice versa. Indeed, the A188V substitution in NS1 of the latest American ZIKV isolates increased viral infectivity and prevalence in mosquitoes. This mutation renders NS1 highly secretable in mammalian hosts, increasing the ability of ZIKV to transmit from host to vector. This may have contributed in part to the recent ZIKV epidemics in South America [77].

A recent study also demonstrated that a mutation in the ZIKV capsid protein (C-T106A) significantly increased the epidemic potential of ZIKV during its transmission cycle by promoting efficient assembly of ZIKV virions. A capsid T106A substitution facilitates ZIKV transmission by the mosquito vector as well as the infection of human cells and immunodeficient mice [78]. Another genome-wide comparative analysis of pre-epidemic ZIKV strains and recently circulating strains suggested that structural changes in the 3′-UTR stem–loop may increase the transmissibility and virulence of ZIKV [79].

Some genetic changes affect virulence, making the virus more contagious or causing more severe symptoms. Mutations in the viral precursor membrane protein (prM) have been shown to exacerbate the symptoms of ZIKV infection. The S139N substitution arose prior to the outbreak in French Polynesia in 2013 and has been present in the subsequently circulating American strains. This mutation may increase the infectivity of neural progenitor cells and promote apoptosis, possibly contributing to microcephaly and other conditions in pregnant women in the recent epidemic [80]. Another substitution of the envelope protein (E-V473M) that occurred before ZIKV spread to the Americas in 2013 has been shown to increase neurovirulence, maternal-to-fetal transmission and viremia, thereby promoting urban transmission. While this mutation did not affect oral infection by *Aedes aegypti*, competition experiments in cynomolgus macaques showed that this mutation increased fitness for viremia production [81].

### 3.3. Japanese Encephalitis Virus

Phylogenetic studies suggest that JEV originally occurs in the Indonesia/Malaysia region and then spreads northward, despite its first detection in Japan [82]. Based on the nucleotide sequence of the envelope protein gene, JEV is currently classified into five different genotypes: GI, GII, GIII, GIV and GV [82]. GIII was responsible for many Japanese encephalitis epidemics and was the most commonly detected genotype in Asia before the 1990s [83,84]. GI was first isolated in Cambodia in 1967 and consists of two clades, GI-a and GI-b. GI-b gradually replaced GIII in the 1990s, and became the most common cause of Japanese encephalitis outbreaks in this region [85].

In order to identify the genetic determinants underlying the GI/GIII lineage replacement, the genome and envelope protein sequences of GI and GIII strains were subjected to in silico analysis. Interestingly, GI was shown to be neutral, while GIII was predicted to have a greater positive selection advantage [83]. A possible explanation is that the lower diversity of GI strains suggests less vector diversity and more efficient replication and transmission cycle between mosquitoes and pigs. This hypothesis was further validated in mosquito C6/36 cell lines, in which GI strains had a significant replicative fitness advantage over GIII strains [86]. Another study analyzing the sequence of three GI isolates from the Midnapur district, West Bengal, in India found a common N103K substitution in the envelope protein in all these GI isolates and suggested that it might be crucial for the GIII to GI replacement and escaping of GIII vaccination [87]. Analysis of viral replication in avian and mosquito cells showed that from 24 to 48 h post-infection, a GI-b JE-91 isolate was significantly more infective in mosquito cells than the GI-a and GIII isolates, suggesting the GI-b isolates had enhanced replication capacity in early phases of mosquito infection and shortened extrinsic incubation period compared to the GIII isolates, leading to subsequent displacement of the GIII lineage [88].

### 3.4. West Nile Virus

Additional examples of viral genetic variation associated with increased arbovirus infectivity, vector fitness and transmissibility have been reported in West Nile virus. The amino acid substitution T249P in the NS3 helicase of North American WNV increased its virulence in American crow, the main natural reservoir of WNV [89]. On the other hand, the replacement of V159A in the envelope protein of the NA/WN02 strain, which replaced the original New York NY99 strain in 2002 as the predominant WNV strain in North America, shortened the extrinsic incubation period in *Culex* mosquitoes, contributing to the prevalence of WNV among mosquitoes. These mutations, in combination, may have led to the rapid spread and persistence of WNV in North America [90,91].

### 3.5. Chikungunya Virus

CHIKV has been repeatedly introduced into Asia from Africa since the 18th century, leading to the establishment of the endemic Asian CHIKV genotype and, in 2005, the Indian Ocean lineage (IOL) [92,93]. During the 2006–2010 epidemic, *A. aegypti* was the main vector of CHIKV in India and other countries. The substitutions K211E in envelope protein E1 and V264A in E2 have been reported to enhance CHIKV adaptation to *A. aegypti* [94,95]. During the recent 2016 CHIKV outbreak in Brazil, researchers identified two new mutations in the virus, K211T in E1 and V156A in E2, that enhanced mosquito adaptations [96].

The Indian Ocean lineage originated from an East/Central/South African (ECSA) enzootic genotype that appeared in 2004 in coastal Kenya and subsequently spread to Indian Ocean islands, India, Southeast Asia and Europe [92,97]. The genetic adaptation of IOL strains to the novel urban vector *Aedes albopictus* via an E1-A226V substitution appears to have contributed, at least in part, to the evolutionary success of this emerging lineage [97,98]. This E1-A226V substitution is responsible for increased midgut infectivity, dissemination and transmission by *A. albopictus* but has little effect on *A. aegypti* infection [98,99]. Epidemiological and phylogenetic studies have shown that the endemic Asian strains of CHIKV that circulate in areas where both *A. aegypti* and *A. albopictus* are present have not adapted to *A. albopictus*. Instead, recent outbreaks in Asia are attributed to introduced ECSA strains that are better adapted to *A. albopictus* [100,101].

The adaptation of CHIKV to *A. albopictus* appears to be a multistep process. Several second-stage adaptive mutations have been identified that also increase CHIKV fitness in this vector [102]. The E2-L210Q substitution was first identified in IOL strains circulating in south India in 2009. This mutation increases the ability of CHIKV to infect and disseminate in *A. albopictus* by a factor of 4 to 5 [102,103]. Together, these mutations may be responsible for the rapid spread of CHIKV by increasing the prevalence of CHIKV in *A. albopictus*, a vector that inhabits Southeast Asia, and by facilitating the spread of the virus to urban centers and areas with cooler climates.

## 4. Conclusions and Perspectives

Vector-borne infectious diseases impose a significant burden on global public health. It is extremely difficult to predict which mosquito-borne virus will be affected, as well as the time and location of the next emerging epidemic, because outbreaks depend on a variety of factors. While factors such as climate, transportation, human and vector population density and vector capacity are important, genetic variations in the virus are the least predictable. Genetic studies of viruses isolated from mosquitoes and patients have made great progress in understanding the patterns of transmission and disease. Continued advances in the efficiency, speed and cost of DNA sequencing now provide new opportunities to revisit many aspects of viral epidemiology and evolution. For example, next-generation sequencing technologies can efficiently sequence large numbers of RNA virus genomes from a single host to provide more complete information about quasispecies, rather than just identifying consensus sequences from RT–PCR amplicons. This will provide a more thorough understanding of the population dynamics and size of the virus at different stages of infection and transmission and may also lead to the identification of sequence variants that exist as minority groups.

Understanding the forces that shaped the evolution of mosquito-borne viruses and the role these forces play in shifting infection to higher virulence is important for developing effective countermeasures, especially in the absence of licensed vaccines or antiviral therapies to control the spread of viral pandemics. Undoubtedly, there are as yet undiscovered mosquito-borne viruses that have the potential to emerge, and recent advances in deep sequencing offer new opportunities to identify them based on surveillance activities, especially in tropical regions with the highest viral diversity. The ability of RNA viruses to rapidly mutate and exploit new ecological opportunities highlights the threat of emerging or re-emerging mosquito-borne viral diseases. As humans continue to invade and alter habitats where mosquito-borne viruses spread, especially through deforestation and urbanization, there will be increasing selection pressure for the continued adaption of these viruses. Thus, retrospective and experimental studies of viral evolution, host range and adaptation are needed to improve our ability to predict disease emergence and to develop new intervention strategies.

## Data Availability

Not applicable.

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
