# Peer review of "Adaptive Evolution as a Driving Force of the Emergence and Re-Emergence of Mosquito-Borne Viral Diseases"

_viruses, 2022, doi:10.3390/v14020435_

Round 1
Reviewer 1 Report
I read with pleasure and interest the work of colleagues Yu and Cheng. I am not able to say if it is sufficiently exhaustive or not but the topic is extremely actual, I also believe that an active surveillance system cannot ignore the monitoring of the evolution of the viral genome. For this reason, I think could be helpful to make possible the dissemination of this work.
Reviewer 2 Report
The reviewer has some comments as below.
-The authors should mention more about ADE when they talk about DENV pathogenicity.
-The authors should add Japanese encephalitis virus, since it has been evolved and genotype shifting has occurred recently. It will be a valuable information as a review article.
-Since all the viruses talked in this review are mosquito-borne viruses, maybe it’s better to change the title from “…of Atboviral diseases” to “mosquito-borne viral diseases.”
Line 61: The sentence “To date, no vaccine candidate…” should be transferred to the end of this paragraph.
